# Toxic Epidermal Necrolysis: A Clinical and Therapeutic Review

**Gonçalo Canhão** [1,*] [ID]**, Susana Pinheiro** [2] **and Luís Cabral** [2,*]

1 Faculty of Medicine, University of Coimbra, 3000-548 Coimbra, Portugal
2 Department of Plastic Surgery and Burns, Centro Hospitalar e Universitário de Coimbra, 3000-075 Coimbra, Portugal
* Correspondence: goncalo.canhao.98@hotmail.com (G.C.); jlacabral@gmail.com (L.C.)

**Abstract:** Toxic Epidermal Necrolysis is a rare dermatological condition with high mortality and serious consequences on its survivors. Despite having been first described in 1956, its pathophysiology remains uncertain, mainly regarding its mechanisms, although it seems that certain apoptosis pathways are pivotal in starting keratinocytes' apoptosis and in activating T cells, especially those mediated by tumour necrosis factor, Fas-FasL and granulysin. In general, its aetiology and presentation are consensual, being defined as a generalized necrolysis of the epidermis that occurs as an uncontrolled immune response to a specific drug or one of its metabolites, highlighting cotrimoxazole and allopurinol as the most important. This necrolysis leads to a massive shedding of the epidermal layer of the skin, with stronger incidences in the torso, upper limbs and face. Its complications tend to be severe, noting that septic ones are responsible for over half of the disease's mortality. Nearly all survivors develop long-term sequelae, namely hypertrophic scarring and skin pigmentation anomalies. Regarding treatment, many different opinions arise, including contradictory ones, regarding more importantly immunomodulation therapies that have been the focus of several studies through the years. It is safe to state that supportive therapy is the only modality that has significantly strong evidence backing its efficacy in reducing mortality and improving prognosis, which have improved in the past years as general health care quality increased. In conclusion, it is imperative to say that more research is needed for new potential therapies with large study populations and more scientific rigor. Likewise, investigation towards its basic pathophysiology should also be promoted, mainly at a biomolecular level, allowing for an improved prevention of this illness.

**Keywords:** toxic epidermal necrolysis; Lyell's syndrome; ALDEN; TNF; IVIG; SCORTEN

## 1. Introduction

Toxic Epidermal Necrolysis (TEN), also known as Lyell's Syndrome, is a rare dermatological condition of great clinical severity [1–3]. It was described for the first time in 1956 by Alan Lyell [4], after whom it was named, and in most cases, it derives from the exposure to certain drugs, accounting for 1% of all hospitalizations for adverse drug reactions [1]. Regarding its clinical features, TEN is characterized by generalized mucocutaneous necrolysis, with bullous lesions and epidermal detachment affecting more than 30% of total body surface area (TBSA) [3,5,6]. This syndrome is rare, with an annual incidence of 1–2 cases per 1,000,000 people [1]. Its mortality is rather high, varying from 25% to 35% [5] but reaching 50–70% [2] depending on sources. TEN's pathophysiology is not fully understood, and there are several theories suggesting autoimmune mechanisms that may lead to keratinocyte apoptosis and necrosis. However, it is known that this immune response is cell-mediated, namely, by T cells [1,2,6,7]. A few theories propose cellular apoptosis mechanisms to be involved, especially those of Tumour Necrosis Factor (TNF), Fas-FasL and granzymes such as granulysin [1,6–8]. In the literature, it is consensual that, in order to achieve the best therapeutic conditions and to ensure the maximum survival rates, patients should be preferentially admitted and treated in Burns Units. As of writing this article, and despite several new treatment modalities having been studied and proposed, particularly

immunomodulation, the only treatment for which its efficacy has been proven and that is widely used is the one built on general support care. However, studies on these new treatments need larger and more representative populations in order to be statistically significant [1,3,7,9–11]. TEN is an important study subject, with considerable potential for new discoveries. With this article, the authors mean to review and summarize information and scientific evidence available. During the study, not only converging points amongst several authors were noted but many interesting targets for research were also referred regarding its pathophysiology and its therapeutic approach. There are still some controversies and disagreements that need better clarification in order to optimize patients' management and outcome.

## 2. Materials and Methods

This study searched for articles and studies about Toxic Epidermal Necrolysis and SJS/TEN overlap syndrome that had been published, in English, between 1 January 2001 and 31 December 2021 using the online databases PubMed and MEDLINE with the search terms "Toxic Epidermal Necrolysis", "TEN", "Lyell's Syndrome", "SJS/TEN", "ALDEN", "SCORTEN" and "IVIG". After the selection of 40 articles and a careful read of their abstracts, 12 articles were excluded and only 28 were considered for inclusion in the study, which covered TEN and SJS/TEN overlap syndrome. This review included 1 systematic review with metanalysis, 11 literature reviews, 2 guidelines, 1 national protocol, 3 prospective original articles, 9 retrospective original articles and 1 case-report study. All that did not meet the aforementioned criteria were excluded from this review, such as a single case case-report and an expert opinion article, as well as 10 others that focused mainly on Stevens–Johnson Syndrome. Each paper included was thoroughly read, highlighting and comparing relevant entries and flagging points of agreement and of disagreement amongst authors.

## 3. Discussion

Toxic Epidermal Necrolysis, or Lyell's Syndrome, is a rare dermatological condition of great severity [1–3] integrating a nosological spectrum with Stevens–Johnson's Syndrome (SJS) and SJS/TEN overlap syndrome [3,5]. These three syndromes are characterized by a sudden onset of high fever, extensive mucocutaneous necrolysis and systemic toxicity, mainly as a response to exposure to certain drugs or pharmacological groups [1,2]. Some authors consider SJS and TEN to be a single entity given their similarities and their difficult discrimination, called SJS/TEN [6]. Out of academic interest, it was defined that cases of epidermal necrolysis affecting less than 10% of TBSA would be considered SJS and those affecting more than 30% would be considered TEN, situating the SJS/TEN overlap syndrome between them [3,5].

### 3.1. Epidemiology

TEN has a global annual incidence of 1–2 cases per 1,000,000 people. Its mean mortality ranges from 25% to 35%, possibly reaching 50–75% if it is not correctly managed [1,2,5,6]. This syndrome represents about 1% of all hospitalizations for adverse drug effects [1]. TEN has a 1.000-times-higher incidence in HIV-positive individuals, reaching a global annual incidence of 1 case per 1.000 people in these patients [3]. Although it is not restricted to any specific age group, it is more common at age extremes: before 5 years and after 64 years of age. It also affects more women than men in a proportion of 2:1 or even 3:1 [1].

### 3.2. Aetiology

In most cases (80–85%), the origin of TEN is tethered to an idiosyncratic reaction to a dose-independent exposure to certain pharmacological groups [1,2,6], but it is important to note that there is a small percentage of patients that develop TEN by unknown non-pharmacological mechanisms [12]. Over 220 drugs have been linked to TEN, with higher or lower frequencies [3,5]. As there is no trustworthy test that conclusively proves a specific

drug's causality, it is safer to speak only of a suspect or probable drug as the causal agent [3]. To evaluate the risk of occurrence of TEN, an Algorithm of Drug Causality in Epidermal Necrolysis (ALDEN) was created, for which its use has been validated as a reference tool [6] (Table 1).

**Table 1.** Algorithm of Drug Causality in Epidermal Necrolysis (ALDEN) [12].

| Criteria | Value | | Rules of Application |
|---|---|---|---|
| Latency between drug administration and onset of symptoms (index day) | Suggestive | +3 | From 5 to 28 days. |
| | Compatible | +2 | From 29 to 56 days. |
| | Probable | +1 | From 1 to 4 days. |
| | Improbable | −1 | More than 56 days. |
| | Excluded | −3 | Drug administered on index day. |
| | NOTE: If there is a previous reaction to the same drug, it is considered "suggestive +3" from 1 to 4 days and "probable +1" from 5 to 56 days. | | |
| Probability that the drug was present in the patient's system | Definitive | +0 | Drug administered until index day or stopped less than 5 elimination half-lives before index day. |
| | Doubtful | −1 | Drug stopped more than 5 elimination half-lives before index day, with abnormal renal and/or hepatic functions or suspected pharmacological interactions. |
| | Excluded | −3 | Drug stopped more than 5 elimination half-lives before index day, with normal renal and hepatic functions and no pharmacological interactions. |
| Prechallenge or rechallenge | Positive specifically for disease and drug | +4 | Occurrence of SJS/TEN [1] after the use of the same drug. |
| | Positive specifically either for disease or drug | +2 | Occurrence of SJS/TEN after the use of a similar drug or another adverse reaction to the same drug. |
| | Positive non-specifically | +1 | Occurrence of another adverse drug reaction to a similar drug. |
| | Unknown/not performed | +0 | No knowledge of previous exposure to the drug. |
| | Negative | −2 | Previous exposure to the drug without any adverse reaction of any kind. |
| Dechallenge | Neutral | +0 | Drug stopped or unknown. |
| | Negative | −2 | Drug not stopped without worsening of clinical condition. |
| Drug notoriety | Strongly associated | +3 | High risk drug. |
| | Associated | +2 | Lower but proven risk drug. |
| | Suspect | +1 | Ambiguous epidemiology; drug "under surveillance". |
| | Unknown | +0 | All other drugs, including new ones. |
| | Not suspect | −1 | No evidence of association. |
| INTERMEDIATE SCORE = −11 to +10 | | | Sum of all previous criteria. |
| Other possible aetiologies for the symptoms? | Possible | −1 | List all other administered drugs according to their intermediate score and if at least one > 3. |

[1] Stevens-Johnson Syndrome/Toxic Epidermal Necrolysis.

This algorithm allows an individual assessment of drug causality, potentially reducing treatment costs and informing the patient about which drug is contraindicated from then on. This algorithm is specific for epidermal necrolysis, encompassing SJS and TEN, and measures six parameters: (1) time spent between drug intake and onset of reaction (index day); (2) probability that the drug was present in the patient's system at the onset of reaction; (3) prechallenge or rechallenge, which are the occurrence of any adverse reactions to a prior administration or a subsequent administration of that specific drug; (4) dechallenge, i.e., an improvement of the patients clinical status after the drug's removal; (5) drug notoriety; and (6) other possible etiological alternatives. ALDEN should be calculated for every single drug known to be taken by the patient, ranging from −12 to +10. Its application distributes drugs into five different categories (Table 2): very probable (≥6); probable (4 to 5); possible (2 to 3); unlikely (0 to 1); or very unlikely (<0) [12].

**Table 2.** ALDEN score interpretation [12].

| Final Score | Classification |
|:---:|:---:|
| <0 | Very unlikely |
| 0–1 | Unlikely |
| 2–3 | Possible |
| 4–5 | Probable |
| ≥6 | Very probable |

Amongst the many pharmacological groups that have been linked to TEN (Table 3), the most frequent are the following: sulphonamides, especially cotrimoxazole, that represent nearly 33% of all cases in adults; antiepileptics such as phenytoin, the most frequent in paediatric ages, carbamazepine and phenobarbital; allopurinol; oral penicillin; non-steroid anti-inflammatory drugs (NSAIDs) with long half-life, namely pyrazolone and the oxicam group; and, more recently, nevirapine and lamotrigine [1,3,5–7,9].

**Table 3.** Highrisk drugs for the development of TEN [3,5,6,13].

| Pharmacological Group | Strong Association | Less Strong Association | Weak Association |
|---|---|---|---|
| Antibiotics | Sulphonamides (especially cotrimoxazole) | Amoxicillin<br><br>Cephalosporines<br>Macrolides<br>Quinolones (typically, ciprofloxacin)<br>Tetracyclines | Nitrofurantoin<br><br>Vancomycin |
| Antiepileptics | Carbamazepine (mainly in paediatric ages)<br>Lamotrigine<br>Phenobarbital<br>Phenytoin | | Valproate |
| Analgesic and anti-inflammatory drugs | NSAIDs [1] from 'oxicam' group | Acetic acid NSAIDs (e.g., diclofenac)<br>Ibuprofen | Acetaminophen (paracetamol)<br>Tramadol |
| Antidepressants | | Sertraline | Fluoxetine<br>Mirtazapine |
| Other | Allopurinol<br>Sulfasalazine<br>Nevirapine | Pantoprazol | Diltiazem |

[1] Non-steroidal anti-inflammatory drugs.

However, there are confounding factors that may impact the identification of a causal drug. For example, oral penicillin, acetaminophen (paracetamol) and corticosteroids are usually administered to treat non-specific symptoms that can be premature ones of TEN [12,13]. The risk of development of TEN is usually limited to the first two months of treatment [7,12], and the first three weeks are the most critical [3,9]. For this reason, it is considered that a previous exposure to a specific drug with no adverse reactions diminishes the probability of that drug being the culprit [12].

An individual who develops TEN by exposure to a certain pharmacological group does not necessarily have an increased risk of developing it to another. Furthermore, a reaction to a specific drug, such as a sulphonamide antibiotic, does not mean that the patient will react to another sulphonamide, such as diuretics, furosemide or cyclooxygenase-2 inhibitors [7]. In the literature, several other risk factors have been identified, namely bacterial infections by *Klebsiella pneumoniae*, *Mycoplasma pneumoniae* or *Yersinia enterocolitica*; vaccinations, especially for measles, mumps and rubeola, for hepatitis B, for chickenpox,

for Influenza virus and for *Haemophilus influenzae* B; allogenic bone marrow transplants and stem-cell transplants; systemic lupus erythematous (SLE); and radiotherapy and some oncological diseases [1,9]. Nonetheless, cases related to vaccines, chemical substances and fumigants are extremely rare and remain as an exception to the rule [7]. As said previously, patients with active HIV infection are 1000 times more likely to develop TEN. This could be attributed to three factors: the higher number of drugs administered, the immune system's qualitative deficiency or abnormal patterns of metabolization of antiretroviral drugs [9]. As such, before prescribing any high risk drug to a seropositive patient, the potential risk of TEN should be kept in mind [3]. Despite an association to the mutation HLA-B*1502 reported in Asian populations that favours the development of TEN by carbamazepine [3,5], no such correlation was found between this or any other mutation and TEN in the remaining populations [5].

### 3.3. Pathophysiology

TEN remains an important study target given that its pathophysiology is not fully understood. The scientific community agrees that, in its core, there is an immunological mechanism mediated by cytotoxic T cells. These cells are the most frequently found in inflammatory infiltrates of desquamative areas and in bullous fluids [1,2,6]. T cells have, in fact, a granulysin-mediated cytolysis mechanism [6,8]. However, the reasons why this immune system's unregulated response occurs are still uncertain. The main mechanism is the massive apoptosis of keratinocytes supported by the identification of several receptors of apoptosis pathways on their cell membranes, more importantly the TNF, Fas-FasL and TRAIL groups [2]. This apoptosis process is mediated essentially by Fas-FasL and perforin-granzyme B [8,14], which can be overactivated by the presence of a specific drug or one of its metabolites. The response is then amplified by inflammatory cells, mainly CD$^{8+}$ T cells and soluble inflammatory mediators [14]. It is thought that TNF-$\alpha$, which can be secreted by macrophages and by keratinocytes, may have a fundamental role in TEN, either by recruiting cytotoxic cells or by inducing the apoptosis of keratinocytes. This molecule was identified in epidermal samples of TEN patients and also in fluids collected from bullous lesions and peripheral blood. Notwithstanding, TNF's role is not yet clarified as it could have a proapoptotic or antiapoptotic effect in TEN. A few studies have shown that thalidomide (an immunomodulating anti-TNF drug) may have a deleterious effect on these patients, increasing their mortality. This seemingly supports the theory that TNF could play an anti-apoptotic role in TEN. Objectively, increases in TNF and FasL levels were observed in TEN patients, but these mechanisms are unspecific and present in other pathologies, thus not explaining why some patients develop TEN and suggesting that there may be a rare polymorphism that alters their functions in controlling apoptosis [2].

Another accepted theory is known as "p-i Concept". It is based on a direct interaction between drugs and class I Major Histocompatibility Complexes (MHCs) that triggers hypersensitivity reactions mediated by CD$^{8+}$ T cells, with granulysin-controlled cytolysis [6]. According to this theory, the culprit drug sets in motion an immune response mediated by MHC, with the clonal expansion of CD$^{8+}$ T cells and interleukin-2 (IL-2) secretion, followed by keratinocyte apoptosis that can occur in two phases: one guided by T cells, as it happens in other dermatological adverse drug reactions, and that is highly dependent on granulysin and cellular death pathways; and another with response amplification that is specific to TEN [2].

Granulysin is a cytolytic protein that may have a key role on TEN's pathophysiology. Its levels in skin biopsy samples surpassed all other cytolytic proteins, such as granzyme-B, perforin and FasL [8]. Additionally, its mechanism appears to be specific to SJS/TEN, and its levels are directly linked to clinical severity [8,15]. The secretion of granulysin in high levels by T cells, Natural Killer (NK) cells and Natural Killer T cells (NKT) leads to undue apoptosis and tissue damage, which appears to culminate in these patients' typical clinical presentation. Granulysin also operates as a chemotactic agent and activates pro-inflammatory molecules. High concentrations of this molecule in the extracellular space of

necrotic and bullous lesions are a probable cause of the rapid development of epidermal necrolysis observed in TEN [8]. It is known that IL-15 increases granulysin secretion [15], and it is possible that granulysin is potentiated by the remaining cytotoxic molecules, causing a synergic effect that worsens keratinocyte apoptosis. Measuring granulysin levels in liquid collected from bullous lesions may be a useful tool in differential diagnosis and an important biomarker for evaluating disease progression [8].

The implication of drug metabolism in TEN is not clear; however, some metabolites, namely hydroxylamine derived from sulphonamide or aromatic antiepileptics, quickly bind to cells if they are not properly removed by epoxide hydroxylase. These metabolites become antigenic when displayed on cell surfaces and can activate apoptosis pathways [6].

In spite of the extreme rarity of a second episode of TEN, the observation of a reduction in latency between drug exposure and clinical onset in a recurrence (from 12–14 days to only 2 days) suggests that there may be a primary sensitivity mechanism and immunological memory [6]. In fact, it is relatively common that TEN survivors develop autoimmune diseases such as Systemic Lupus Erythematous (SLE) or Sjögren's Syndrome [1,6].

### 3.4. Clinical Presentation

Drug exposure is usually followed by a prodromal period, with unspecific symptoms (fever, myalgia, malaise, anorexia and asthenia) or even rhinitis, cough and chest pain. These symptoms can last between 1 and 14 days [1,2,5,6]. The first mucocutaneous symptoms begin to appear abruptly 2 to 3 days after the prodromal period, initiating the acute phase, which can last from 2 to 12 days. Generalized pruritus is commonly the first manifestation, but it is rapidly followed by painful eruptions that appear mainly in the face and torso, despite possibly spreading centrifugally towards the remaining body parts in just a few days. The most frequently affected areas are the torso and proximal upper limbs [1]. The initial lesions are erythematous macules, with irregular borders and a darker central region (target-like), reaching its maximum size in 2 to 3 days depending on the culprit drug's half-life [1,6,13,14]. These macular lesions coalesce and quickly turn into bullous ones, with clear fluid, creating great plaques of necrotic epidermis. Keratinocyte necrosis takes place essentially in the spinous and basal layers of the epidermis, which detaches cleanly off the dermal layer and remains intact albeit exposed [1,6]. The loss of the epidermis is accompanied by the detachment of fingernails and a loss of eyebrows [15]. At supposedly healthy areas, a slight smear pressure may trigger the shedding of the epidermis; this is called Nikolsky's sign, which is an important tool for differential diagnoses [1–3,5,6,13]. A triad can be defined for TEN comprising mucosal eruptions, epidermal necrosis with desquamation and target-like lesions [5]. Epidermal necrolysis can involve the entire body, generally sparing the scalp [1,7], and the total loss of the epidermis in less than 24 h is not uncommon [3]. Desquamative areas are identifiable by their exudative and dark-red dermis. Epidermal detachment leads to fluid, protein and electrolyte losses, similarly to burn patients, and if it is not properly mitigated, it will induce serious hydroelectrolytic and haemodynamic disorders, most importantly dehydration, hypovolemia and acute renal failure [1]. Mucosal lesions are observed in more than 90% of cases and usually precede epidermal necrolysis by 1 to 3 days [3,7,14]. These are mainly erosive, with a loss of conjunctival, oropharyngeal, nasal and/or oesophageal mucosae, or even urethral, anal, vaginal and/or perineal, suggesting a predilection for stratified squamous epithelium [1,6,9]. The extension and the localization of these mucosal lesions is variable and specific to every patient, but they are always painful and can compromise a correct hydration and nutrition routine. Early ocular involvement is extremely relevant, and it is found in almost every patient with TEN and can cause photophobia [1]. Some authors propose that a patient with extensive cutaneous erythema and ocular involvement may safely be assumed as a victim of TEN and, inversely, the absence of ocular involvement almost rules out this diagnosis [9]. Urethral lesions can lead to the urinary retention and necrosis of renal tubules, which, along with hydroelectrolytic disorders, can negatively impact the therapeutic approach to these patients [1,13]. The body's temperature may remain high throughout the entire

acute phase, even without infectious complications. This could be due to the release of endogenous pyrogenic agents by necrotic tissues, especially IL-1 [1].

The most serious complication that frequently leads to death is infection. Sepsis is the main cause of TEN-associated mortality [1,3,6,14,16], accounting for more than 50% and surpassing non-septic multiorgan failure [1,14]. This complication is greatly facilitated by the loss of the epidermal barrier, facilitating the invasion of tissues by bacteria and other microbes from the skin [1,16]. Contrarily to thermic burns, in TEN, the dermis remains intact, although it is still susceptible to invasion by microorganisms that multiply freely in exudates and necrotic epidermis [1]. Cutaneous lesions in TEN, similarly to burns, are primarily colonized by *Staphylococcus aureus* and then followed by Gram-negative bacteria that mainly come from the patient's digestive tract, notably *Pseudomonas aeruginosa* [1,3,16]. Patients under previous treatment with broad spectrum antibiotics can also develop fungal infections, most frequently by *Candida albicans* [1]. Around 25% of patients will suffer from hematological dissemination (bacteraemia) either by *S. aureus*, *P. aeruginosa* or Enterobacteriaceae. Upon the date of admission, there are certain variables that, when present, help clinicians in predicting the risk of bacteraemia and sepsis: age over 40 years-old; leucocytosis over 10.000/mm3; and an affected TBSA of 30% or higher. The identification of colonization by meticilino-resistant *S. aureus* (MRSA) or *P. aeruginosa* in skin cultures is predictive of bacteraemia by the same microorganisms. Its peak incidence is attained after around 11 days after the onset of symptoms or about 5 days after hospitalization [16], and it is greater in patients with central venous catheters (CVC). In some cases, sepsis in TEN can lead to Disseminated Intravascular Coagulation (DIC) [1]. The high prevalence of bacteraemia associated with Enterobacteriaceae strengthens the theory that these patients may experience digestive bacterial translocation [16]. Besides its haematogenic origin, starting from cutaneous lesions or intestinal translocation, sepsis could also develop as a consequence of pneumonia [14].

Multisystemic involvement is relatively common in TEN [9]. Ocular complications are frequent, affecting nearly 74% of patients [3], and they can vary from light conjunctival hyperaemia to the formation of pseudomembranes with the fusion of the eyelid to the ocular globe (symblepharon) which can lead to complete blindness [1,3,6]. These lesions are due to the erosion and desquamation of the conjunctival mucosa, with consequent fibrosis [1,6]. Nevertheless, the most frequent ophthalmological complications are photophobia, xerophthalmy and foreign body sensations [7]. Respiratory disfunctions are a common finding present in 25–30% of patients [3,9], and they can require invasive mechanical ventilation in 10–20%, even without radiographical anomalies [7,14]. Some disturbances can be found through optical bronchofibroscopy. These disfunctions can accrue from several factors, such as superficial breathing caused by pain or pulmonary oedema from increased alveolar–capillary permeability [1]. Additionally, the aspiration of debris of oropharyngeal mucosa can also lead to aspiration pneumonia and bronchiolitis obliterans or even acute respiratory distress syndrome (ARDS) [1,2]. Respiratory difficulties settle in progressively and can go unnoticed, but they can be hinted at by the onset of dyspnoea, tachypnoea and marked hypoxemia [3,7,9]. Its treatment includes saline nebulisations, bronchodilators, respiratory physiotherapy and, when needed, invasive mechanical ventilation. The use of non-invasive mechanical ventilation is not recommended seeing that the pressure and friction associated with facial masks can aggravate perioral and perinasal desquamation. Respiratory failure is a sign of a poor prognosis [2]. Besides oropharyngeal mucosal destruction, gastrointestinal involvement encompasses the appearance of distal erosions, namely at the oesophagus, resembling peptic oesophagitis. These lesions may, in rare occasions, lead to dysphagia and gastric bleeding. Intestinal lesions are less frequent and can be evidenced by haematochezia. Although close to 50% of patients present with rising hepatic transaminases (AST and ALT), only about 10% will develop hepatitis. Haematological disorders are also very common, particularly anaemia, which is usually normocytic and normochromic and could be precipitated by diverse reasons, including erythroblastopaenia. Leukopenia is relatively frequent, with lymphocytopenia occurring in 90% of TEN cases, which can be

explained by the depletion of $CD^{4+}$ T cells; on the other hand, neutropenia is found in 30% of patients and is, generally, associated with sepsis. However, it is uncertain if neutropenia is either caused by medullary disfunction or solely as a secondary idiopathic phenomenon. Thrombocytopenia is the least frequent of the cytopenias and it arises in 15% of cases [1,2].

After its acute phase, the chronic phase begins and this is where long-term sequelae stand out, as they occur in nearly 90% of survivors after 1 year [17]. The most common sequelae are: (1) dermatological, namely dryness of skin, pigmentation anomalies, nail defects, alopecia and alterations of the sudoriferous pattern [1,7,17]; (2) ophthalmological, such as xerophthalmia, cicatrising conjunctivitis, lagophthalmos and symblepharon, which affect visual acuity on various levels, even possibly leading to total blindness; (3) oral, including xerostomia and dental defects; (4) genital, more prominently phimosis in men [7,17]; and, rarely, (5) gastrointestinal and (6) bronchial. It is also important to emphasize the psychological sequelae, mainly post-traumatic stress disorder (PTSD), which can seriously impact the efficacy of future treatments, given the fear of a new TEN [17].

### 3.5. Diagnosis

TEN can be presumed from typical clinical signs, with at least three of the following present: disseminated purpuric maculae or target-shaped lesions; epidermal desquamation; multifocal mucosal erosions; and positive Nikolsky's sign [17]. However, a definitive diagnosis requires a skin biopsy and its histological analysis [1,5,6,17]. The biopsy should be performed as early as possible and also allows the exclusion of differential diagnoses, which benefit from targeted, specific and distinct treatments [7]. The general rule is that two specimens should be collected for anatomopathological analysis: one for routine evaluation with haematoxylin-eosin and another for direct immunofluorescence [5]; there is yet another preparation, Tzanck smear, that can reveal eosinophils and basal cells with an elevated nucleus/cytoplasm ratio [1,4]. From an anatomopathological point of view, the lesions are characterized by total epidermal depth keratinocyte necrosis, with subepithelial bullae and basal membrane vacuolization [1,4,14]. In its initial stages, a predominantly T cell-populated dermal infiltrate is frequently found [3,9,17], but a prompt switch to a macrophage infiltrate can occur [9]. Chung et al. [8] proposed that measuring granulysin levels in bullous lesions' fluid could be an alternative to skin biopsy as a definitive diagnosis exam; however, the latter remains as the gold-standard exam for TEN.

### 3.6. Differential Diagnosis

TEN's differential diagnosis should not only include erythema multiforme (EM), staphylococcal scalded skin syndrome (SSSS), erythema scarlatiniform, toxic shock syndrome, paraneoplastic pemphigus, graft vs. host disease (GVHD) and drug rash with eosinophilia and systemic symptoms (DRESS) syndrome but also other toxic dermatoses, serious dermatological drug adverse effects, autoimmune dermatoses, systemic lupus erythematous, dermatomyositis, thermic and caustic burns and caustic dermatitis [1,3,5,7,9,15,17] (Table 4).

For several years, erythema multiforme was considered to be part of the SJS/TEN spectrum. However, according to pronounced differences between them, currently, it is considered as a distinct nosological entity [3,6,7,9]. EM is characterized by raised lesions that are larger than 3 cm, which may or may not be target-shaped with negative Nikolsky's sign. Usually, lesions keep clear of mucosae and they can be limited to a single body area, affecting, most of the time, less than 20% of TBSA [1,6,9]. Additionally, EM mostly occurs as a late immune response to a specific infectious disease, notably by Herpes Simplex virus (HSV) or *Mycoplasma pneumoniae*. It has a significantly lower mortality than TEN [5,7].

**Table 4.** TEN's main differential diagnoses [1,7,9,17].

| Bullous Disease | Fever | Mucositis | Morphology | Onset | Other Characteristics |
|---|---|---|---|---|---|
| Erythema multiforme | Yes | No | Raised lesions Nikolsky neg. | Late | Lesions after HSV [1] or *M. pneumoniae* infections. |
| SSSS [2] | Yes | No | Painful erythema with perioral crusts | Acute | Mainly in children younger than 5 y-o, but also in immunocompromised individuals or undergoing haemodialysis. |
| Erythema Scarlatiniform | Yes | Yes | Erythema on flexures | Acute | Possible pharyngeal and lingual involvement. |
| Toxic Shock Syndrome | Yes | No | Macular rash on palms and soles, with desquamation | Acute | Multisystemic involvement is more evident. |
| Paraneoplastic Pemphigus | No | Yes, severe | Polymorphic lesions with flaccid bullae | Insidious | Linked to oncological diseases, primarily lymphomas. Refractory to treatment. |
| GVHD [3] | Yes | Yes | Morbilliform erythema, bullae and erosions | Acute | Very similar to TEN. |
| DRESS [4] syndrome | Yes | No | Rash without desquamation | Late | Marked eosinophilia and systemic symptoms. |

[1] Herpes simplex virus; [2] Staphylococcal scalded skin syndrome; [3] Graft versus. Host disease; [4] Drug rash with eosinophilia and systemic symptoms.

Cutaneous infections by *Staphylococcus aureus* may lead to scalded skin-like lesions (SSSS), being frequently misjudged as TEN, despite having an extremely more favourable prognosis and a lower mortality. This syndrome is most frequent in newborns and children, although some cases have been described in immunocompromised adults or undergoing hemodialysis [1,7]. In spite of having a broad spectrum of clinical presentation varying from localized bullae to extensive exfoliation with negative Nikolsky's sign, SSSS does not present itself with painful mucosal nor ocular involvement [1,5,9]. Histological differentiation is also relatively simple seeing that, in SSSS, epidermal necrolysis is only partial, with an intraepidermal detachment of the granular layer and without necrosis of the deepest layers [1,9].

Erythema scarlatiniform is a cutaneous infection caused either by group A β-haemolytic streptococci (such as *Streptococcus pyogenes*) or by *Staphylococcus aureus*, and it can present itself with generalized erythema, which is more marked in flexure areas and with the desquamation of digital pulps, pharyngitis and glossitis [1].

Toxic shock syndrome is caused by *Staphylococcus aureus* and leads to diffuse erythema with desquamation, more pronounced on palms and soles, fever and systemic involvement. This condition rapidly evolves to shock [1].

Paraneoplastic pemphigus manifests as oral mucositis accompanied by generalized polymorphic bullous eruptions, possibly hindering its differentiation from TEN. Nevertheless, this condition is associated with oncological diseases, particularly lymphomas, and its early stages are markedly distinct, with an insidious course of disease and a tendentially chronic and refractory evolution [7].

Graft vs. host disease has a less abrupt onset and it typically spreads in from the extremities to proximal areas, without ocular involvement. The existence of extracutaneous lesions, namely hepatic and gastrointestinal, may help the differentiation from TEN [7,9].

DRESS syndrome is characterized by eosinophilia and systemic symptoms, without epidermal desquamation or mucosal involvement [9,17].

### 3.7. Treatment

3.7.1. General Measures

The most important aspects of TEN's management and treatment are its swift identification, the immediate removal of all non-essential and/or suspect drugs, admission to a burn unit, adequate support therapy, nurse care and multidisciplinary collaboration between colleagues of several medical areas [2,3,17]. It is crucial that healthcare professionals be able to recognize and identify the early signs and symptoms of a possible TEN [17]. As referred, these patients, because of their loss of epidermal barrier, have great difficulties in keeping thermoregulation, resulting not only in discomfort, stress and a catabolic state [13,14,17] but also in increased susceptibility to local and systemic infections, making their isolation in positive-pressure rooms a central matter [1,2]. The prompt transfer to a Burn Unit, which has optimal conditions, technical means and skilled professionals, is of the utmost importance in order to reduce their mortality [1,3,7,9–11,15], which is inversely proportional to the lapse of time between the diagnosis and the admission to this hospital facility [7,10]. Admitting the patient to a Burn Unit effectively reduces infection risk and total time of hospitalization [3]. At the time of admission, it is imperative to gather the patient's clinical history, with special attention to recent administrations of new drugs or exposure to chemical substances; concurrently, a complete and detailed physical exam should also be carried out in order to determine the affected mucocutaneous extension. To calculate the affected TBSA, physicians can resort to systems that are regularly used in burns patients, such as "Rule of 9" or, for greater accuracy, Lund and Browder chart [1]. It is crucial to immediately discontinue all non-essential medicines, including suspect drugs [1,2,9,12,13,17], as the early removal of the suspect drug can diminish the risk of mortality in about 30% per day [14]. An observational study showed that when the culprit drug (with a short half-life) is removed on the same day of the detection of the first cutaneous lesions, mortality decreases from 26% to 5%. However, no such benefit was recorded for drugs with longer half-lives [1,2,7,9]. Essential and non-suspect drugs should be kept on the patients therapeutic table [17]. Blood samples must be gathered in order to conduct a full study, including a complete blood count with leukocyte formula, electrolyte and hepatic tests, infectious parameters and coagulation tests. These analytical analyses should be repeated daily [1,5,7]. It is also important to collect specimens for bacteriological examinations every 48–72 h [17], performing hemocultures, urine cultures, sputum, saliva or tracheobronchial aspirate cultures and swab cultures from desquamative areas [1,5,7,8]. Skin cultures have a low positive predictive value (PPV) and low specificity but an excellent negative predictive value (NPV) of bacteraemia by MRSA or *Pseudomonas aeruginosa*. When it comes to bacteraemia by Enterobacteriaceae, skin cultures' NPV greatly decreases, suggesting that these organisms come not from the skin but from the gastrointestinal tract [16]. The patient's HIV serological status must be checked, as should antinuclear antibodies (ANA) and antibodies to soluble nuclear antigens (SSA and SSB). When determining the suspect drug is not possible when using ALDEN, serological tests and an oropharyngeal swab for *Mycoplasma pneumoniae* and other atypical bacterium (such as *Chlamydia* spp.) should be conducted [17]. If possible, acquiring peripheral venous accesses in non-affected areas is an important step, which allows the administration of fluids [1,17]. Peripheral venous catheters are preferred over central ones because the former are associated with lower infection risk, reserving the latter for exceptional cases and for the minimum time possible [1]. Catheter fixation must be achieved without resorting to adhesive materials [17]. To prevent pulmonary complications, peripheral oxygen saturation should be monitored using regular or continued pulse oximetry and chest X-rays may be useful [1].

At the time, there is no strong or decisive evidence regarding specific treatments for TEN. For this reason, supportive therapies are the most essential step of its management [7,17,18]. The first therapeutic measure is fluid therapy in order to replenish volumes, electrolytes, proteins and glucose levels, ensuring hydroelectrolytic and acid-base homeostasis [3]. Taking the value of affected TBSA, fluid necessities may be calculated resorting to various formulas used in burns patients, namely Parkland's and Brooke's [1]. However,

intravenous fluid and electrolyte necessities must be weighted in a case-by-case basis since TEN patients have less pronounced necessities than burns patients with a similar affected TBSA [1,3,17,19]. Fluid quantities should be calculated to achieve a urinary output of at least 1 mL/kg/h and to correct eventual base deficits, which should be possible with a crystalloid at 2 mL/kg/% of desquamative TBSA in the first 24 h. In the first 72 h, fluid therapy must ensure a urinary output of 0,5 to 1 mL/kg/h, hemoglobin levels of 7 mg/dL or higher and a mean blood pressure of 65–70 mmHg [13,19]. Monitoring urinary output by using urethral catheterization allows for an effective surveillance; however, it should be removed once the patients clinical condition improves [1]. There is no specific reason that favours the administration of colloids over crystalloids seeing that their effectiveness is similar in achieving the aforementioned goals, but the former are usually more expensive and, as such, crystalloids are usually preferred [19]. Nonetheless, if hypalbuminaemia is present, human albumin at 5% may be administered at 1 mL/kg/% of affected TBSA [13]. Patients must be encouraged to maintain adequate oral support for hydration and nutrition [1,13,17], although oropharyngeal and oesophageal lesions can frequently determine the need to insert a nasogastric tube or even to transition to a total parenteral nutrition, which increases the risk of infections [1,2]. Enteral nutrition is always preferable over parenteral nutrition, and it is associated with a lower risk of infection [2]. As in other serious illnesses, hyperglycaemia may increase morbimortality risks and, for that reason, a strict control of capillary glycaemia every 1–2 h is recommended until the acute phase's resolution and every 4 h thenceforth. If two consecutive measures show levels over 180 mg/dL, an insulin infusion scheme should be initiated [17], and according to some authors, insulin can also have an antiapoptotic effect, which is beneficial in TEN [2].

The risk of gastric ulcers imposes a need to prescribe proton-pump inhibitors, especially in patients under parenteral nutrition despite the fact that these drugs facilitate gastric bacterial colonization. This might justify the addition of oral sucralfate [1,13].

Analgesic drugs, primarily opioids, and sedatives are indicated for pain relief, a priority during the acute phase. This is true both for basal pain and for that which arises from therapeutic procedures, which are performed several times a day. In certain clinical moments, it is even required to fall back on general anaesthesia. NSAIDs are not recommended, as they themselves could be implied in TEN's aetiology [1,13,17].

Patient isolation and the adoption of strict aseptic techniques by healthcare professionals are fundamental to prevent the microbial colonization of denuded areas. Balneotherapy under anaesthetic sedation is advocated, carefully removing necrotic cutaneous remains and oral and nasal crusts, and applying topical antiseptics [1]. It is of the most importance to preserve the physiological environment needed for reepithelization and to allow for free mobilization of the limbs. The materials chosen for wound coverage must be durable, comfortable, easy to apply and economically sustainable, and they should not be impermeable, toxic or adherent [3].

Other general measures include anticoagulation with low molecular-weight heparin (LMWH), unless contraindicated [1,13], and daily physical therapy to ascertain limb mobility [3].

### 3.7.2. Wound Management

There are, currently, two schools of thought regarding wound management in TEN: one that relies on more or less invasive surgical care and another that is based on a strongly conservative approach.

If a surgical approach is preferred, some authors recommend a less invasive therapy with surgical debridement of desquamative and wrinkly areas under intravenous sedation, without resorting to skin substitutes for wound covers. This option is only plausible because the dermal layer remains intact, which allows for a perfect reepithelization with only aseptic care of exposed areas using balneotherapy and daily dressings with topical antibiotics until the new epidermal layer is reconstructed. Other authors favour a more invasive approach with a surgical debridement of all desquamative areas, as well as all areas with

positive Nikolsky's sign under general anaesthesia and then applying either biological or synthetic dermatological substitutes for wound coverings. This strategy seems to prevent microbial colonization of exposed areas, reduce pain and associated losses and promote reepithelization [1]. If the more conservative surgical approach is elected, balneotherapy must be performed daily or even twice a day and supplemented with topical chlorhexidine or, preferably, polyhexanide over exfoliative areas, and then dressings with gauze should be used to absorb exudates [1,9,13]. Sulfadiazine is absolutely contraindicated not only because it delays reepithelization, is painful and induces leukopenia but also because sulphonamides are one of the main causal agents of TEN [1,3,9]. A mixture of paraffin may also be applied to the entire epidermis [13]. Regarding dermatological substitutes, biological materials are difficult to obtain and can even be predisposed to local infections. Synthetic materials allow for a reduction in pain, protein loss and local inflammation, and they also accelerate reepithelization and mobility, yet they have no effect over mortality [3]. Biobrane® (Smith & Nephew, Watford, England, UK) is a semisynthetic dermatological substitute that was shown to markedly diminish pain, eliminate the need for further grafts and allow earlier physical therapy [2]. Acticoat® (Smith & Nephew, Watford, England, UK) is a nanocrystalline silver-based dressing that combines antimicrobial activity with anti-inflammatory effects [3]. Aquacel® (Convatec, Reading, England, UK) is a silver hydrofiber that dispenses its removal before skin reepithelization is complete; it is also economically sustainable and reduces local infections, dressing renewal and the pain associated with it [3,20,21]. Aquacel® was also shown to be effective in preventing infections by *Pseudomonas aeruginosa*, *Serratia marcescens*, *Bacteroide fragilis*, *Aspergillus niger*, MRSA and vancomycin-resistant enterococci (VRE) [21]. After surgical debridement, the choice of material heavily depends on the experience of the medical team, the available microbiological exams, the place of treatment, the availability and costs of the product and the seriousness of the disease.

Recently, a new idea has come to light advocating against the surgical debridement of desquamative epidermis [11,17], proposing the use of the desquamative skin itself as a biological cover. This theory suggests that all smear pressures should be avoided to greatly diminish the loss of epidermis—antishear therapy. It showed a greater than expected decrease in mortality both in lower and higher risk patients. Additionally, it allows to prevent sequelae, costs, pain and risks associated with traditional dressings [11].

### 3.7.3. Corticosteroids

The administration of corticosteroids in TEN has been highly controversial. Initially, they were prescribed because it was considered that autoimmunity was the main pathophysiological pathway; however, their immunosuppressing effect potentiates the risk of infectious complications and disguises the early signs of a possible sepsis. They also delay reepithelization and raise mortality [1]. Although, it has been demonstrated that high doses of corticosteroids in the primordial stages of TEN may reduce epidermal loss, as they mitigate the inflammation process [2,5,8,9,21] by inhibiting T cell activation through IL-2 transcription blockage. Some authors suggested the use of intravenous dexamethasone in 1.5 mg/kg/day bolus for three consecutive days [7,10]. On the other hand, other authors raise concerns over the risk of sepsis, longer hospitalizations and higher mortality related to corticoid use [2]. The difficulties in diagnosing TEN in its early non-bullous stages also complicates the use of these drugs [9]. Overall, the scientific community agrees today in not recommending the use of corticosteroids in TEN due to its effect on mortality remaining uncertain and seeing that there are also cases of TEN in patients undergoing therapy with corticosteroids for other reasons.

### 3.7.4. Antibiotics

Prophylactic antibiotic use is discouraged [1,5,7], unless the patient presents with marked leukopenia. In all other instances, antibiotics should be reserved for the first signs of septic complications, such as fever, mental state alterations and/or a sudden onset of

infection/inflammation biomarkers including procalcitonin and C-Reactive Protein (CRP), with a later adjustment of the antibiotics when the antibiotics sensitivity test (AST) is available [1,5,9]. The decision to start antibiotics may be swayed by the fact that early signs of infection are very similar to the ones of a systemic inflammatory response syndrome (SIRS) [16]. At the first sign of sepsis, the empirical prescription of antibiotics is suggested, covering the most frequently involved bacteria in skin cultures (*Staphylococcus aureus* and *Pseudomonas aeruginosa*) and Enterobacteriaceae and according to the local flora and antimicrobial resistance patterns of the Burn Unit [13,18]. It is important to highlight that a massive exudation of fluids through exposed skin areas may lead to a need of greater doses of antibiotics [7]. Systemic fungal infections, mostly by *Candida albicans*, may require the administration of systemic antifungals [1].

### 3.7.5. Prevention and Management of Complications

The high incidence of ocular complications suggests that not only local collyriums are indicated (an antiseptic/antibiotic every hour, a lubricant every 1–2 h and a corticosteroid every 6 h [13] to avoid the accumulation of abrasive crusts over the cornea) but also that daily evaluations by an ophthalmologist should be requested to remove any conjunctival synechia [1,3,5,14]. Amniotic membrane transplant seems to be a promising treatment in reducing ocular sequelae [7,9], as it helps to restore the integrity of corneal epithelium, reduces inflammation and prevents scarring [5,9].

Patients at risk of acute respiratory failure must be promptly intubated and mechanically ventilated. Non-invasive ventilation should be avoided due to facial cutaneous lesions and mucosal desquamation, as referred [17]. Respiratory physical therapy is indicated to prevent pneumonias and atelectasis [1].

Oral washing with a chlorhexidine elixir twice daily helps minimize bacterial colonization of damaged mucosae and maintain a good oral hygiene [13,20]. Additionally, an analgesic elixir every 2–3 h can be offered, as well as a protective one every 6–8 h. Likewise, soft white paraffin should be applied to the lips every 2 h [13].

White soft paraffin should also be applied to urogenital areas every 4 h, accompanied by daily topical corticosteroid in non-desquamative, inflammatory and itchy areas [13].

### 3.7.6. Immunomodulation

Although implying the insertion of a central venous catheter, plasmapheresis is a safe therapeutic option that provides a fast relief of pain and necrolytic activity, also reducing hospitalization times [1]. It has particular interest when initiated in the first days after symptoms occur, as its mechanism of action appears to be the removal or dilution not only of inflammatory mediators but also of the culprit drug and its metabolites [1–3,15,22]. Plasmapheresis should always be considered in the more serious cases of TEN, especially when initial therapeutic measures are ineffective [22].

Other immunomodulating therapies, such as monoclonal antibodies, cytokine inhibitors and others, have been proposed, but they are not widely accepted or used given the absence of robust studies proving their efficacy and safety.

N-acetyl cysteine (NAC), frequently used as a mucolytic agent or as an antidote to acute acetaminophen intoxication, has shown to be effective in high-doses (up to 1 g, 6 id) when treating TEN. Several reasons may explain this. It is possible that it is related to its support on cellular antioxidant activities. It could be attributed to the increase in intracellular levels of cysteine, which is necessary for the production of glutathione. Finally, it may inhibit the production of cytokines that mediate the immune response, such as TNF-α and IL-1, as well as oxygen-free radicals [1,9].

Pentoxifylline is a peripheral vasodilator that may be beneficial in TEN as it interferes with T cell's link to keratinocytes and with cytokine production (namely TNF-α, IL-1 and IL-6), both from macrophages and from keratinocytes [1,7].

Granulocyte colony-stimulating factors (G-CSF) may be useful to circumvent neutropenia associated with TEN, reducing the risk of sepsis [1,2]. In fact, the daily use of

5 µg/kg of filgrastim in most severe cases promotes recuperation and reepithelization, regardless of neutrophil count [13].

In in vitro studies, zinc has been shown to be capable of protecting cells against chemically, physically or immunologically (including pharmacological) induced apoptosis. Zinc also works as an antioxidant and cell membrane stabilizing agent. Zinc supplements of elemental zinc 150 mg, 2 id, for six weeks can have good results in TEN, inhibiting keratinocyte apoptosis and, in sufficient doses, acting as an immunosuppressant [2].

Given the similarities between TEN and GVHD, as well as the mechanisms involved, the use of immunosuppressing drugs can be indicated. Among them, cyclosporin stands out. It inhibits T cells, macrophages and inflammatory cytokines, preventing keratinocyte apoptosis [1,2]. Seeing that the use of immunosuppressants always carries heavy risks, low doses are recommended: one of 3–5 mg/kg/day [2,7]. Unfortunately, when compared to support therapies, cyclosporin did not significantly improve reepithelization or morbimortality [9], and it remains as an important study and debate subject without consensus regarding its empirical use [13,17]. In the same line of thought, cyclophosphamide was brought up as an alternative. However, its toxicity is not specific to immunological cells and it can even worsen keratinocyte apoptosis [2]. For these reasons, and despite having studies that prove its efficacy, it is not possible to come to a conclusion on its role in TEN and, as such, it remains out of the therapeutic options [7,13].

Anti-TNF agents have also been suggested. Thalidomide inhibits TNF and IL-6 production, but its use is contraindicated as it was associated with an increase in mortality [2,13]. On the other hand, infliximab and etanercept may help reduce inflammation [15]. More studies with bigger population groups are needed to prove anti-TNF agents' efficacy and safety in TEN.

Intravenous immunoglobulin G (IVIG) therapy has been a topic of focus in the past years, and it remains controversial. Several studies have been conducted on this thematic, evaluating the patients' responses. Theoretically, IVIG may be beneficial in eliminating the culprit drug, in inhibiting FasL associated mechanisms and in clearing necrolytic mediators [2,23], reaching maximum utility in the first 72 h after bullous lesions appear [20]. A dose of 1 g/kg/day during three consecutive days is recommended [7,14,23–25]. This dose has shown to be safe only with mild adverse effects, which typically occur in the first 30–60 min of treatment and are self-limited; the most common side effects are headaches, myalgia, fever, nausea and vomiting [25]. IVIG has permitted a reduction in mortality [10]. Some authors suggest the use of IVIG in combination with other therapies, namely plasmapheresis [3,23] or corticosteroids [5,21]. Despite possessing a lower toxicity than other immunomodulating therapies and a relatively low risk of serious side effects [7,26,27], the global scientific community remains reluctant in incorporating this therapy in their recommendations due to the lack of sufficiently strong evidence that may support its expensive use [5,9,25]. Some authors contradict the supporting studies, claiming they have unsurpassable biases [18] or that results are incoherent and unclear [13,15]. Other authors report studies in which no clear improvements in morbimortality were achieved after treatments with IVIG [17].

Table 5 presents a summary of all aforementioned therapeutic options in TEN.

**Table 5.** A summary of therapeutic options in Toxic Epidermal Necrolysis.

| | Therapeutic Options | Use and Recommendation | Notes |
|---|---|---|---|
| General measures | Fluid therapy<br>Proton-pump inhibitors | 1st line therapy.<br>Recommended. | At 2 mL/kg/%TBSA to ensure a UO ≥ 1 mL/kg/h.<br>For the prevention of gastric ulcers. |
| | Analgesics | Recommended. | For basal and procedure-induced pain.<br>General anaesthesia may be needed. |
| | LMWH [1] | Recommended, unless contraindicated. | For the prevention of thromboembolic events. |
| Corticosteroids | | Not recommended.<br>Can be useful in primordial stages of TEN. | A daily bolus of dexamethasone 1.5 mg/kg for 3 consecutive days may help reduce inflammation. |
| Antibiotics | | Recommended, if clinical or analytical signs of infection are present. | Prophylactic use is strongly discouraged.<br>The empirical choice must cover for *S. aureus*, *P. aeruginosa* and Enterobacteriaceae.<br>Adjust antibiotics to local flora and resistance patterns of the Burn Unit. |
| Immunomodulation | Plasmapheresis | Recommended, in serious cases or if ineffective initial therapies. | Maximum usefulness in the first days of symptoms. |
| | NAC [2] | Can be considered. | Till 1 g, 6 id. |
| | Pentoxifylline | Can be considered. | |
| | G-CSF [3] | Can be considered. | Can be useful to minimise neutropenia.<br>Filgrastim 5 μg/kg, 1 id. |
| | Zinc supplements | Can be considered. | At 150 mg, 2 id, for 6 weeks. |
| | Cyclosporin | Needs further clarification. | Empirical use not recommended, but at 3–5 mg/kg/day could help reduce inflammation. |
| | Cyclophosphamide | Not recommended. | |
| | Anti-TNF [4] Thalidomide | Absolutely contraindicated. | Associated with higher mortality. |
| | Infliximab Etanercept | Needs further clarification. | |
| | IVIG [5] | Needs further and more thorough investigation. | It is safe at 1 g/kg/day for 3 consecutive days, but its efficacy is uncertain. |

[1] Low molecular weight heparin; [2] N-acetyl cysteine; [3] Granulocyte-colony stimulating factor; [4] Tumour necrosis factor; [5] Intravenous immunoglobulin.

### 3.8. Prognosis

TEN can be considered a self-limited disease that, in ideal conditions and in the absence of complications, may resolve without any sequelae. Reepithelization begins during the acute phase and can last from 1 to 3 weeks, but mucosae require more time to heal. It is expected that high fever persists until the complete resolution of the case, even without infectious complications [1,5]. Most chronic sequelae are dermatological (81–100%), including abnormal scarring and nail dyschromia and dystrophia, ophthalmological (27–54%), oral (12,5%) and vulvovaginal, gravely endangering the survivors' quality of life [3,9]. Patients' evolution is dependent on clinical and laboratorial factors, most of them directly associated with a worse prognosis. Among them, we can highlight delays in the removal of non-essential drugs and in transferring the patient to a Burn Unit, age, basal status and prolonged reepithelization time (over 9 days). Analytically, persistent neutropenia is the condition most frequently associated with mortality, reflecting the patient's lower capability of resisting to infectious phenomena [1].

The risk of death is seriously increased by the presence of complications, which were included in the Severity of Illness Score for Toxic Epidermal Necrolysis (SCORTEN; Table 6). This score should be calculated in the first 24 h of admission and again on the third day of hospitalization [9,14,15]. The computed variables are as follows: age over 40 years; active oncological disease; heart rate over 120 beats per minute; initial affected TBSA over 10%; blood urea nitrogen (BUN) over 28 mg/dL; glycaemia over 252 mg/dL; and serum bicarbonate under 20 mEq/L [2,5–7,9,13–15,17,28]. Each parameter is accounted for as either '0' or '1'.

**Table 6.** SCORTEN [2,5–7,9,13–15,17,28].

| Parameter | Reference Value |
|---|---|
| Age | >40 years. |
| Oncological disease | Yes. |
| Desquamative TBSA | >10%. |
| Heart rate | >120 bpm. |
| BUN | >28 mg/dL. |
| Glycaemia | >252 mg/dL |
| Serum bicarbonate | <20 mEq/L. |

After adding the scores, an estimated mortality is attributed to the value of SCORTEN (Table 7).

**Table 7.** SCORTEN associated mortality [6,9].

| SCORTEN | Estimated Mortality (%) |
|---|---|
| 0–1 | 3.2 |
| 2 | 12.1 |
| 3 | 35.3 |
| 4 | 58.3 |
| >5 | 90.0 |

As it was developed base on healthcare data from 1979 to 1998, some authors contested its present-day accuracy by suggesting an overestimation of mortality. However, a recent metanalysis has demonstrated no significant differences between SCORTEN's estimated mortality and real mortality [28]. SCORTEN has indeed shown to be exceptionally precise in predicting mortality in TEN. Notwithstanding, a review of its parameters may be useful, adding more precise parameters when it comes to age and TBSA. Additionally, it would be of relevance to embody new ones, such as delay in hospitalization, previous therapies with corticosteroids or antibiotics, thrombocytopenia or leukopenia and renal disfunctions [7,28].

## 4. Conclusions

TEN remains a rare and extremely serious illness for which its early diagnosis is limited by the unspecific features of its early symptoms. However, a quick transfer to a Burn Unit and the removal of all non-essential drugs are two simple but life-saving measures that greatly improve patients' outcomes. For that matter, it is of the utmost importance that healthcare professionals are trained to recognize and identify possible early signs of TEN.

Even though its clinical aspects are well characterised, there is still plenty to uncover regarding its pathophysiology. However, significant progresses have been made, as several protagonists identified, over the past decades, namely, granulysin, Fas-Fasl, TNF and T cells. This a great area for research, with real possibilities for the discovery of new fundamental pathways that may change how we are able to respond and, overall, prevent TEN.

Despite many treatment modalities have been tested and debated, especially in the field of immunomodulation, it must be stressed that isolation, infection control and support therapy still prevail over every one of them, remaining as the only ones universally recommended. In the literature, there is a great controversial debate about the efficacy and security of these new therapies, and the need for more rigorous studies and multicentred clinical trials is evident, with more representative population samples, in order to achieve an optimal approach to a disease with such a high morbimortality.

**Author Contributions:** Conceptualization, G.C. and L.C.; methodology, G.C.; validation, G.C., S.P. and L.C.; formal analysis, S.P. and L.C.; investigation, L.C.; resources, G.C. and L.C.; data curation, L.C.; writing—original draft preparation, G.C.; writing—review and editing, G.C., S.P. and L.C.; visualization, G.C.; supervision, S.P. and L.C. All authors have read and agreed to the published version of the manuscript.

**Funding:** This research received no external funding.

**Institutional Review Board Statement:** Not applicable.

**Informed Consent Statement:** Not applicable.

**Data Availability Statement:** No new data were created or analyzed in this study. Data sharing is not applicable to this article.

**Conflicts of Interest:** The authors declare no conflict of interest.

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
