# Peer review of "Toxic Epidermal Necrolysis: A Clinical and Therapeutic Review"

_2673-1991, doi:10.3390/ebj3030036_

Round 1
Reviewer 1 Report
Canhao et al. present data on clinical and therapeutic responses of toxic epidermal necrolyis covering and evaluating pertinent current literature.
Following comments:
Only 28 articles are included, what are the inclusion and exclusion criteria?
Evidence levels should be given.
line 175: the sentence on thalidomide is not clear. Why should anti-apoptotic effects be more present?
Line 206: what does „consenual“ mean in this context?
Line 296: „respiratory difficulties settle in insiduously“?
Line 332: why perform DIF? For differential diagnosis? HE stainded cryostat sections may be prioritized by excellerating technical procedures.
Line 355: lesions are rather larger than bigger.
Line 426: antibodies to soluble nuclear antigesn.
Line 440: what is basilar stone?
Line 444: please give a citation for Parkland and Brooke.
line 453: explain colloids over crystalloids.
Lne 476: what is hydrotherapy?
Line 513, line 519: was showed effective? Was shown to be effective?
line 608: what doses of zinc? Oral application?
Line 695: „a cloud of doubt still aires over ist pathophysiology“ rather epic, please reword.
Alopurinol reads allopurinol.
Line 98 and other places: „isn’t“ better read „is not“
Interesting and comprehensive paper, yet more summarizing than critically evaluating.
Evidence levels should be given.
Reviewer 2 Report
The authors performed a review on the TEN/SJS. It is well written and interesting. The paper is a little bit lengthly. I would also suggest to add one table summarizing current treatment options for TEN/SJS - it would facilitate the reading of the paper.
Reviewer 3 Report
on the substance, the text is very poor and does not provide any new information. It is rough or even sometimes incomprehensible in certain paragraphs such as physiopathology. On the form, it is not pleasant to read because there are perpetual redundancies between text and tables. There are constant repetitions. The style is rather low level. Conclusions: to be reviewed with significant modifications or to be rejected
Reviewer 4 Report
Congratulations to the authors for a very thorough analysis of the existing literature about TEN and for a critical approach to the topic. A very important advantage of the manuscript is to organize the knowledge on the subject in a synthetic and understandable way for the reader.It is worth mentioning that every doctor is obliged to report adverse drug reactions to the appropriate authorities in a given country.
Round 2
Reviewer 1 Report
The manuscript has considerably improved.
A list of abbrevations should be included, especially for new table 5.